# A Rare Case of *Rhizomucor pusillus* Infection in a 3-Year-Old Child with Acute Lymphoblastic Leukemia, Presenting with Lung and Brain Abscesses—Case Report

**DOI:** 10.3390/idr18010002

**Published:** 2025-12-23

**Authors:** Yanko Pahnev, Boryana Avramova, Natalia Gabrovska, Yolin Dontcheva, Genoveva Tacheva, Krasimir Minkin, Hans Kreipe, Nadezhda Yurukova, Marin Penkov, Nikola Kartulev, Zdravka Antonova, Velichka Oparanova, Nadezhda Tolekova, Petia Moutaftchieva, Bogdan Mladenov, Plamena Hristova, Kaloyan Gabrovski, Svetlana Velizarova, Albena Spasova, Hristo Shivachev

**Affiliations:** 1Department of Pediatric Surgery, University Multiprofile Hospital for Active Treatment and Emergency Medicine ‘N.I.Pirogov’, 1606 Sofia, Bulgaria; n.kartulev@gmail.com (N.K.); zantonova1978@abv.bg (Z.A.); oparanova@abv.bg (V.O.); nadejda.tolekova@gmail.com (N.T.); petiamout@gmail.com (P.M.); hshivachev@gmail.com (H.S.); 2Department for Pediatric Hematology and Oncology, University Hospital ‘Quin Joanna’, 1527 Sofia, Bulgaria; b.avramova@isul.eu (B.A.); jolin.doncheva@gmail.com (Y.D.); n.yurukova@isul.eu (N.Y.); mpenkov77@gmail.com (M.P.); 3Department of Pediatrics, Specialized Hospital for Active Treatment of Children’s Diseases “Prof. Ivan Mitev”, Medical University-Sofia, 1606 Sofia, Bulgaria; natalski@abv.bg (N.G.); sv_velizarova@abv.bg (S.V.); doc_spasova@abv.bg (A.S.); 4Children Neurology Unit, Department of Pediatric Neurology, University Pediatrics Hospital, Medical University-Sofia, 1000 Sofia, Bulgaria; gtacheva@medfac.mu-sofia.bg; 5Department of Neurosurgery, University Hospital St. Ivan Rilski, 1407 Sofia, Bulgaria; minkin@abv.bg (K.M.); k_gabrovski@abv.bg (K.G.); 6Institute of Pathology, Medical School Hannover, 30559 Hanover, Germany; kreipe.hans@mh-hannover.de; 7Department of Pediatric Anesthesiology and Intensive Care, University Multiprofile Hospital for Active Treatment and Emergency Medicine ‘N.I.Pirogov’, 1606 Sofia, Bulgaria; bogdanmladenov@gmail.com (B.M.); h_plamena@yahoo.com (P.H.)

**Keywords:** *Rhizomucor pusillus*, complicated *Mucormycosis*, lung abscess, brain abscess, acute lymphoblastic leukemia

## Abstract

Invasive *Mucormycosis (IM)* is an extremely rare infection with a high mortality rate, caused by a group of fungi classified as *Mucorales* moulds. *Rhizomucor pusillus* is a saprophitic, thermophilic, and angioinvasive microorganism that grows and lives at about 45 °C and is usually found in different environmental spaces such as soil, air, water, food, and other organic matter. These features predispose the infection to wide dissemination, especially in immunocompromised patients and most often in children after chemotherapy for hematological malignancies (HMs). *Mucormycosis* in patients with hematologic malignancies and neutropenia represents between 0.07% and 4.29% of the concomitant diseases. *IM* can develop into an infection in different sites, but its most common manifestation is pulmonary, followed by rhino-orbital–cerebral and disseminated forms. In recent years, an increased morbidity rate has been associated with the ongoing *COVID-19* pandemic, as cited in the literature. There are many publications with *COVID-19-associated mucormycosis* (CAM) cases. The present treatment protocol includes extensive and radical surgical debridement and systemic antifungal therapy with Liposomal Amphotericin B (L-AmB), Posaconazole, and Isavuconazole, either combined or as monotherapy. Despite these new treatment modalities, the mortality rate remains over 50%. We present a rare case of a 3-year-old child with acute lymphoblastic leukemia (ALL) and systemic *Rhizomucor pusillus* infection, diagnosed on the occasion of lung and brain abscesses. The patient underwent lung and brain surgery and is recovering well with no further complications. The two-year follow-up period shows no signs of recurrence of the disease.

## 1. Introduction

Invasive *Mucormycosis* (IM) is an extremely rare infection with a high mortality rate, caused by a group of fungi classified as *Mucorales* moulds. In the literature, the first publication on the subject dates back to 1945. The most commonly found genera in the human population are *Rhizopus*, *Mucor*, and *Rhizomucor* [1]. *Cunninghamella*, *Absidia*, *Saksenaea,* and *Apophysomyces* are rarely isolated [1,2,3,4]. *Rhizomucor pusillus* is a saprophitic, thermophilic, and angioinvasive microorganism that grows and lives at about 45 °C and is usually found in different environmental spaces such as soil, air, water, food, and other organic matter [1,3,4]. These features predispose the infection to wide dissemination, especially in immunocompromised patients and most often in children after chemotherapy for hematological malignancies (HM). *Mucormycosis* in patients with hematologic malignancies and neutropenia represents between 0.07% and 4.29% of the concomitant diseases [5,6,7]. IM can develop into an infection in different sites, but its most common manifestation is pulmonary, followed by rhino-orbital–cerebral and disseminated forms [7]. In recent years, an increased morbidity rate has been associated with the ongoing *COVID-19* pandemic, as cited in the literature [7]. There are many publications with *COVID-19-associated mucormycosis* (*CAM*) cases. The present treatment protocol includes extensive and radical surgical debridement and systemic antifungal therapy with Liposomal B (L-AmB), Posaconazole, and Isavuconazole, either combined or as monotherapy. Despite these new treatment modalities, the mortality rate remains over 50% [8].

We present a rare case of a child with acute lymphoblastic leukemia (ALL) and systemic *Rhizomucor pusillus*, diagnosed on the occasion of lung and brain abscesses.

## 2. Case Report

A 3-year-old male was diagnosed with ALL, and treatment was initiated for him according to the ALL-IC-BMF 2009 protocol, with induction using Prednisone, Vincristine, Doxorubicin, and Pegaspargase at the Department of Pediatric Hematology and Oncology at the University Hospital in Sofia, Bulgaria. After the first two weeks of chemotherapy, on day 15, the patient was proven to be in remission (bone marrow MRD—1.86%), with an MRD of 0% on day 33. One month after the onset of leukemia, during the chemotherapy-induced bone marrow aplasia (with 12 consecutive days of neutropenia), a febrile state with a body temperature of up to 39 °C and a mild cough was registered. The chest X-ray showed discrete paratracheal inflammatory shadowing in the right upper pulmonary lobe. An antimycotic (Voriconazole), antibiotics (Meropenem and Teicoplanin), and G-CSF therapy were initiated. One week later, a chest X-ray was performed again due to the worsening of the patient’s condition. The results showed pneumonia in the right upper lobe (Figure 1).

The next day, the patient presented with a generalized seizure, without a febrile state. The condition was treated with Diazepam. The child presented no other neurological symptoms after that incident and was transferred to a tertiary pediatric hospital for further diagnosis. On day 40, after the initiation of chemotherapy, during his admission, he had a positive test for *COVID-19*, but the patient was asymptomatic. The following three control *COVID-19* nasopharyngeal swabs, performed on a daily basis, were negative. An anticonvulsant therapy with intravenous Phenobarbital was started. On the MRI of the central nervous system and body CT scan, a left occipital ischemic stroke and right upper lobe pneumonia were found. (Figure 2).

An active *Tuberculosis* (TBC) disease was excluded using both the Interferon gamma-release assays (IGRAs)—the T-SPOT.TB and QuantiFERON-TBgoldPlus tests. The antibiotic therapy and Voriconazole were continued. In the first blood culture test, *Candida* spp. and *Coagulase*-*negative* Staphylococci (*CoNS*) were confirmed, and the therapy was switched to Levofloxacin, Teicoplanin, and Anidulafungin. An anticoagulant therapy with monitoring of the levels of Antithrombin III and Factor Xa was performed. The patient’s condition improved over the next few days, and he tested negative for *COVID-19* infection on RT-PCR.

On day 14 from the onset of symptoms, a chest CT scan was repeated due to persistent fever and deterioration of the patient’s general state (Figure 3). The results showed pulmonary abscess in the right upper lobe. (Figure 4).

On the same day, the child was admitted to the pediatric thoracic surgery department for further treatment of the advanced pulmonary complication. The patient was febrile and intoxicated but without signs of respiratory insufficiency and with an SpO_2_ level of 94–96%. No shortness of breath, tachypnoea, or wheezing was registered. Bronchial breath sounds were presented in the right apical pulmonary segments. Ultrasound showed moderate hepatomegaly. The neurological status showed generalized muscle weakness and opsomyoclonus symptoms with no meningeal syndrome. Antibiotic, anticonvulsant, and anticoagulant therapies were continued. Laboratory tests showed a persistent inflammatory constellation. In the following blood sample, *CoNS/MRS* and *Enterococcus faecium* were isolated. Both bacteria were sensitive only to Vancomycin, Linezolid, and Teicoplanin. The antibiotic therapy was changed to Meropenem, Vancomycin, and Anidulafungin. The central venous lines of the patient were not used for diagnostic or therapeutic purposes to prevent further septic episodes. No positive blood cultures or PCR were obtained from the removed catheters.

An ultrasound-guided percutaneous fine-needle aspiration of the lung abscess was performed on day 3 after admission. Despite the proper positioning of the needle, no fluid was evacuated, and the patient was transferred to the operating theatre for video-assisted thoracic surgery (VATS). A 5 mm two-port VATS excisional abscessotomy was performed (Figure 5).

The abscess cavity was filled with necrotic tissue, but no pus or other extensive liquid collection was found. There was no significant pleural effusion. Two chest tube drainages were positioned in the pleural and abscess cavities separately. Postoperatively, the patient manifested with partial and total atelectasis of the right lung (Figure 6), while no air or fluid leakage through the drainages was noticed. Daily bronchoscopies were performed accordingly.

The microbiological evaluation of the bronchoalveolar lavage and the pleural exudate was negative. The methodologies used for the diagnosis were cultures, staining, and the BioFire^®^FilmArray^®^ PCR system. The histological specimen acquired from the affected pulmonary tissue was positive for TTF-1, LCA, CD-3, CD-20, CD-68 + Gomori Grocott, and PAS, suggesting, but non-significantly, mycotic colonies. The microbiological test showed *Aspergillosis*, but less than 50 copies per millilitre. Due to the delayed healing process, deterioration in the patient’s general condition, high fever, and persistent total right lung atelectasis on the 5th postoperative day, a CT scan and an MRT scan were performed. The results showed the enlargement of the inflammatory zone in the right upper pulmonary lobe and a round hypodense lesion approximately 30 mm in diameter in the left occipital supratentorial region (Figure 7 and Figure 8).

The lesion in the left occipital lobe was hyperintense on T2 and displayed a moderate restriction on diffusion-weighted imaging. The differential diagnosis was between ischemic stroke and cerebritis. An extended multidisciplinary consilium was conducted. A decision to perform radical lung resection was made in accordance with the protocols for the treatment of life-threatening mycotic infections, resistant to systemic therapy. The brain abscess was also discussed as a possible mycotic etiology. A right posteriolateral thoracotomy with right upper lobectomy was performed (Figure 9).

The special stains (Grocott, PAS-diastase) showed sparse, irregularly contoured, broad mycotic hyphae up to 3 µm. The histopathological specimen was evaluated by a reference at the Department of Pathology in Hannover, Germany, for a second opinion. The result was the following: “Removed lung tissue revealed extensive and coalescent necrosis, demarcated by lung tissue with scarring reaction and type 2 proliferates. Bronchi also contained granulocytic exudate. In the necrosis and in the exudate within the bronchi branching microorganism with rather slender hyphae and not as broad irregularly contoured and pleomorphic as in typical *Mucormycosis* could be detected in sparse numbers (inset upper right corner, Grocot stain). In the molecular analysis of DNA extracted from the paraffin material PCR amplification was performed with consensus primers from the ITS region and hybridization was done with the VisonArray Fungi chip 1.0 (Zytovision, Hannover, Germany). The hybridization chip covers 30 clinically relevant fungi specimens including *Candida*, *Aspergillus*, *Cryptococcus neoformans*, *Fusarium* spp., *Lichtheimia corymbifera*, *Mucor* spp., *Paecilomyces variotii*, *Pneumocystis jirovecii*, *Puerpruecillium lilacinum*, *Rhizomucor pusilius*, *Rhizopus* spp., *Scedosporium* spp., *Trichophyton/microsporum*. A positive hybridization result was obtained with *Rhizomucor pusillus* DNA.” (Figure 10).

The final diagnosis was established as mycotic granulomatous chronic pneumonia with extensive necrosis, associated with *Rhizomucor pusillus*.

On the third postoperative day after thoracotomy, the antimycotic therapy was changed to Liposomal Amphothericin B and considered for a prolonged treatment for at least 8–12 weeks. In the first 3 postoperative days, the child still manifested total atelectasis of the right lung. Daily bronchoscopies were performed, and bronchoalveolar lavages were conducted. On postoperative day 4, the right lung was totally re-expanded, and the pulmonary function improved. On day 23 after his admission, the patient was discharged afebrile from the surgical department with no neurological symptoms and in complete ALL remission. The treatment continued in the Department of Neurosurgery. The control MRI performed one month after the first MRI revealed enlargement of the left occipital lesion and ring enhancement, which corresponded to the diagnosis of brain abscess (Figure 11).

The neurological examination was normal, considering the impossibility of examining the visual fields at this age. The enlargement of the brain lesion despite the antimycotic treatment with L-AmB was considered an indication for surgical treatment. Craniotomy with complete abscess evacuation and partial capsula resection because of the involvement of the visual cortex and pathways was performed (Figure 12).

The postoperative course was uneventful. The MRI performed 40 days after the neurosurgical operation demonstrated a complete shrinkage of the abscess cavity (Figure 13).

The microbiological evaluation using BioFire^®^FilmArray^®^ PCR confirmed the same results as in the specimens from the thoracotomy and was later confirmed using DNA hybridization.

After, the patient was transferred back to the Department of Pediatric Hematology and Oncology, where a monotherapy of three courses of Blinatumomab was administered.

One year after the diagnosis of the mycotic infection, the control full-body CT scan showed that the patient was in clinical remission, with no evidence of the leukemia or relapse of the disease in the brain or lung (Figure 14).

The systemic treatment for *Rhizomucor pusillus* with Amphothericin B-PEG at a dose of 5 mg/kg daily was administered for six full months. On the third month after the onset of the Amphothericin-B therapy, Posaconazole was added as a concomitant drug at a dose of 200 mg, administered three times per day for eight months. We adapted the doses of antimycotics, and the disease was under control. Posaconazole was switched to Isavuconazole on the tenth month after the onset of antimycotic therapy for six months at a dose of 8 mg/kg to overcome the drug resistance. The patient tolerated the long antimycotic treatment only with rare episodes of hypokalemia as a side effect. The overall duration of the antimycotic therapy was sixteen full months.

The follow-up period was eventless, and the patient was stable, with no febrile state and a lack of any neurological or respiratory symptoms. The control blood tests showed no inflammatory activity and/or positive inflammatory markers such as CRP and procalcitonin.

## 3. Discussion

IM is a life-threatening infection with a high mortality rate, especially in children with HMs. These patients have a poorer prognosis than those with other comorbidities [8]. A constantly increasing incidence is cited by different authors.

The total number of citations found by the time of the aforementioned issue on different search engines and scientific communities concerning *Rhizomucor pusillus* was 178, of which 17 discuss the infection in pediatric populations. Only four of them are systematic reviews, and one is a meta-analysis of the general population. Most of the conclusions and recommendations are based on forensic materials and reports.

According to an extensive European study, the morbidity of IM increased from 1.2 per 100,000 inpatients in the period from 1988 to 2006 to 3.3 per 100,000 inpatients from 2007 to 2015, of which 52.6% had HMs [9]. Another large prospective multicenter study in North America and Japan showed that 75 (56.39%) of 133 patients and 74 (61.2%) of 122 patients diagnosed with IM were also suffering from HMs [7,10,11,12]. This data proves that HMs are the most common risk factor for IM. Few articles find that acute leukemia, neutropenia, and steroid therapy are among the highest risk factors for IM in patients with HMs [7]. An Australian study from 2016 shows that in 36 patients with HMs and *Mucormycosis*, 4 (11.1%) were positive for *Rhizomucor pusillus* [13].

The manifestation of the disease depends on the anatomical site of the infection, and it is classified into six types: pulmonary, rhino-orbital–cerebral, soft-tissue, gastrointestinal, disseminated, and other rare forms (cardiac, renal, osteomyelitis, etc.). Among these, the most common in the pediatric population are the pulmonary and disseminated forms [7,14]. A European trial from 2011 analyzes 102 patients diagnosed with HMs and IM registered in the European Confederation of Medical Mycology. The results show that the major anatomical sites of infection are pulmonary (35.34%) and disseminated (28.27%) [15]. Another cumulative result from 11 different studies and 608 reviewed patients revealed that the most common form of IM in patients with HMs is pulmonary (44% of all cases). The most common symptom is fever, varying in wide degrees. The pulmonary form often presents with dyspnea, cough, and chest pain [7]. Visual diagnostics show exudation, cavity, ground-glass lesion, consolidation, pleural effusion, halo sign, atelectasis, reverse halo sign, etc. [7,16].

The clinical presentation in our case is a result of two localizations—pulmonary and cerebral—with unspecific pulmonary symptoms and seizures. Pneumonia is not an unusual complication in children with leukemia, but its localization in the upper lobe of the right lung is more typical of TBC rather than bacterial pneumonia. Also, the characteristic of this pneumonia (purulent, with very quick formation of necrosis and atelectasis), in spite of the aggressive antibiotic treatment, is suspicious of mycotic infection.

The prognosis of *Mucormycosis* is poor, with a high mortality rate in patients with HMs. A comparative study of 70 patients, divided into two groups, shows that if Amphotericin B therapy is initiated up to the 6th day after the onset of symptoms, the mortality rate is twice as low as in the group that started the treatment more than 6 days after the onset of symptoms [17]. Muthu et al. published a meta-analysis of 1544 patients in 2021, showing that disseminated *Mucormycosis* has a significantly higher risk of fatal outcomes, compared to isolated PM [18]. The results also prove that the multimodal treatment, including extensive surgery and medical therapy, affords a better survival rate than conservative therapy alone [30.9% (43/139) vs. 69.3% (228/329)] [18].

The delayed diagnosis of the disease is due to the biology of the microorganism and the specificity of the infection. In addition, there is a lack of a standardized protocol for diagnosis and treatment. The key to the successful management of the disseminated form of IM with a low mortality rate is its early, accurate diagnosis and prompt radical surgical treatment. The diagnostic gold standard is the histopathological and microbiological evaluation of the specimen and culture. The quantitative PCR/DNA sequencing proves the diagnosis in most of the cases [7,19,20]. Some authors advise a combination of PCR and high-resolution melt analysis (PCR/HRMA), especially for *Rhizomucor pusillus* and *Lichtheimia* and other *Mucor* spp. in patients with HMs and IM [21].

Lately, plasma cell-free DNA next-generation sequencing (NGS) is becoming more popular and proves to be very effective in detecting *Mucorales*. Xu et al. published results from a retrospective comparative study of 347 patients with HMs and IM infection, showing that plasma cell-free DNA NGS is twice as accurate as all the other culture tests (72.6% vs. 31.4%, *p* < 0.001) [22]. In our case, we confirm the diagnosis only with molecular genetic testing of the biopsy specimen because the standard diagnostic methods for mycosis were unsuccessful.

According to a recently published review of guidelines for diagnosing and treating *Mucormycoses*, surgical management is the first line of therapy. In their guidelines, the European Confederation of Medical Mycology (ECMM) and the European Conference on Infections in Leukemia–6 (ECIL-6) recommend that radical resection with clear surgical margins is crucial for the outcome of the treatment [7].

In two major consecutive publications, Zaoutis et al. and Roilides et al. describe 246 pediatric patients with *Zygomycoses*, and most successful treatment and good results were achieved with the combination of amphotericin B and radical surgical resections [23,24]. A multicenter retrospective analysis of 39 patients with IM and HMs in Israel reported decreased mortality in a group of 26 cases treated with extensive surgical procedures compared to another 13 treated only with medications (16% vs. 71%, *p* < 0.001) [25]. Another retrospective multicenter study of 74 patients with IM in 16 intensive care units (ICUs) in France, for the period 2008–2017, reported a better survival rate in patients in intensive care units (ICUs) treated with combined surgical debridement and antimycotic therapy. The authors found that surgical therapy in their cases had a close connection with the survival rate in patients with HMs and IM (OR = 0.71, *p* < 0.001) [26].

Various publications prove that surgical treatment improves the disease’s overall survival rate and contributes to a low mortality rate, but only in patients with well-controlled underlying disease [7,27].

In our case, there were two challenges—the first was to confirm the diagnosis, and the second was to choose the right antifungal treatment.

## 4. Conclusions

*Rhizomucor pusillus*, as part of *Mucormycosis*, is associated with high morbidity and mortality rates in pediatric patients with hematological malignancies. Early diagnosis, adequate systemic antifungal therapy, and radical surgery are the keys to successful treatment of disseminated *Mucormycosis*. The diagnosis and treatment of this serous complication require an extensive multidisciplinary team and highly specialized clinical, microbiological, pathological, and genetic laboratories with experience with this diagnosis. Despite the good results prior to the initial treatment, the patient should be under close follow-up in case of any late complications or the dissemination of the infection. The success of the treatment depends on its early start, using the most relevant antimycotic drugs and a long enough therapeutic course.

## Figures and Tables

**Figure 1 idr-18-00002-f001:**
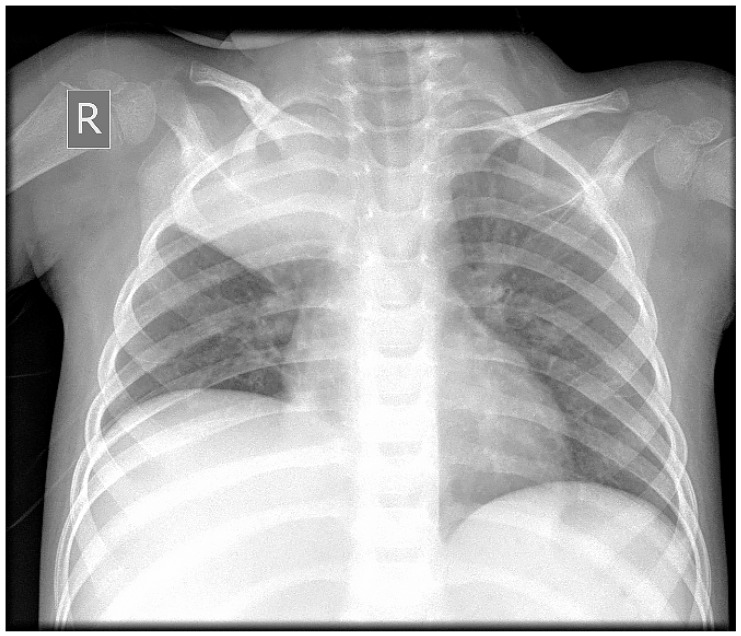
X-ray showing right upper lobe pneumonia (R—X-ray positive tag, marking patient’s right side).

**Figure 2 idr-18-00002-f002:**
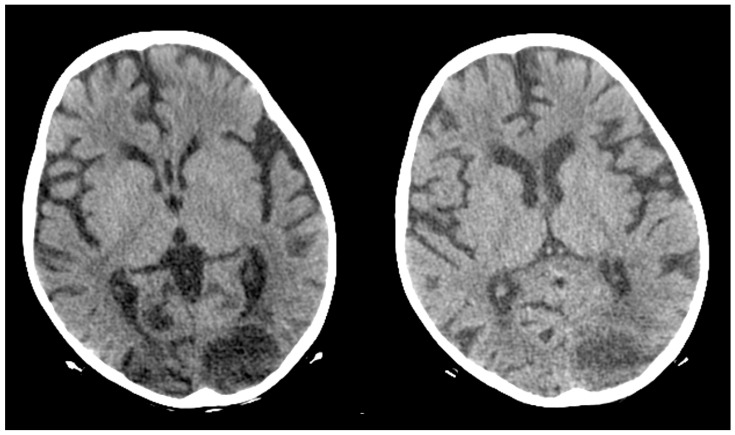
CT scan showing left occipital ischaemic stroke or abscess.

**Figure 3 idr-18-00002-f003:**
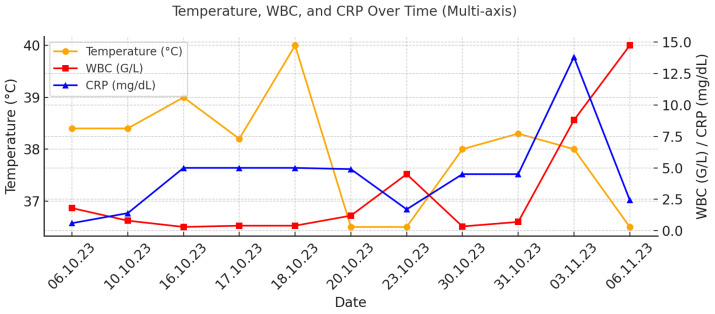
Time distribution diagram demonstrating the progression of the main clinical and laboratory inflammatory markers—fever, white blood cell count, and C-reactive protein.

**Figure 4 idr-18-00002-f004:**
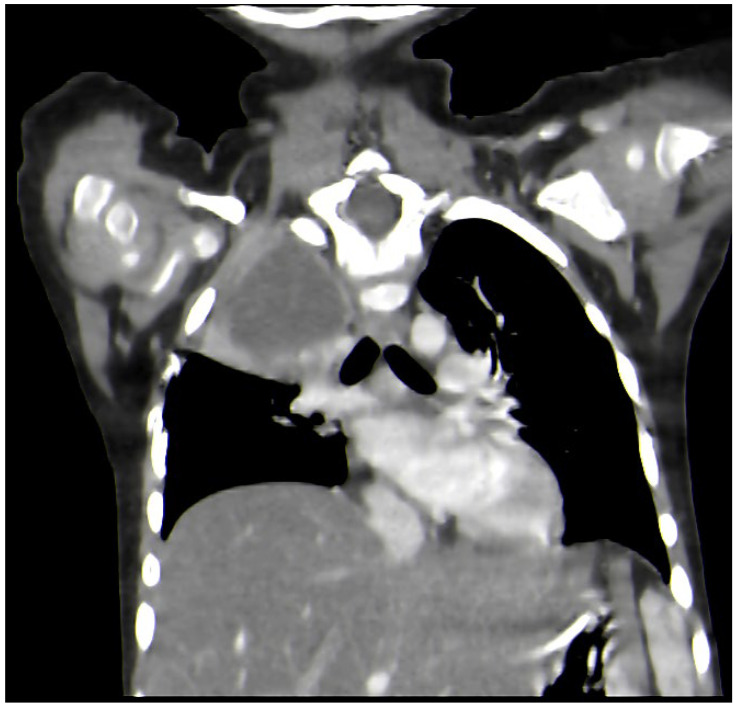
Lung X-ray on the day of admission, showing right upper lobe pulmonary abscess.

**Figure 5 idr-18-00002-f005:**
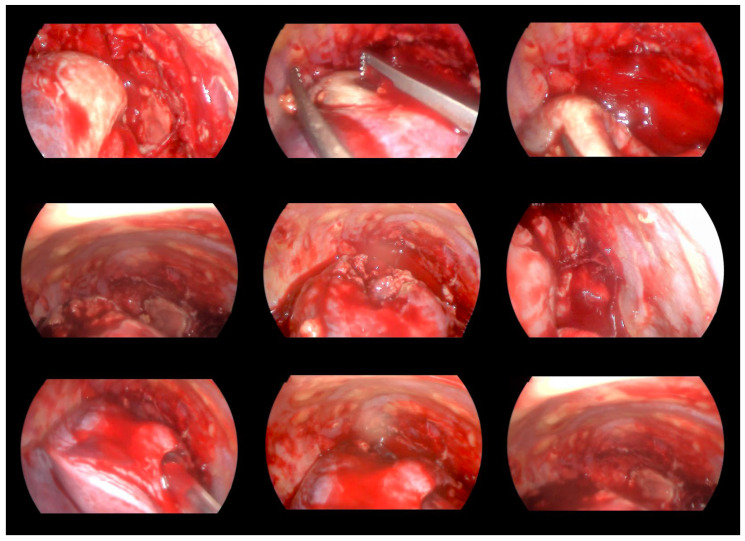
VATS. Abscessotomy with target drainage of the cavity.

**Figure 6 idr-18-00002-f006:**
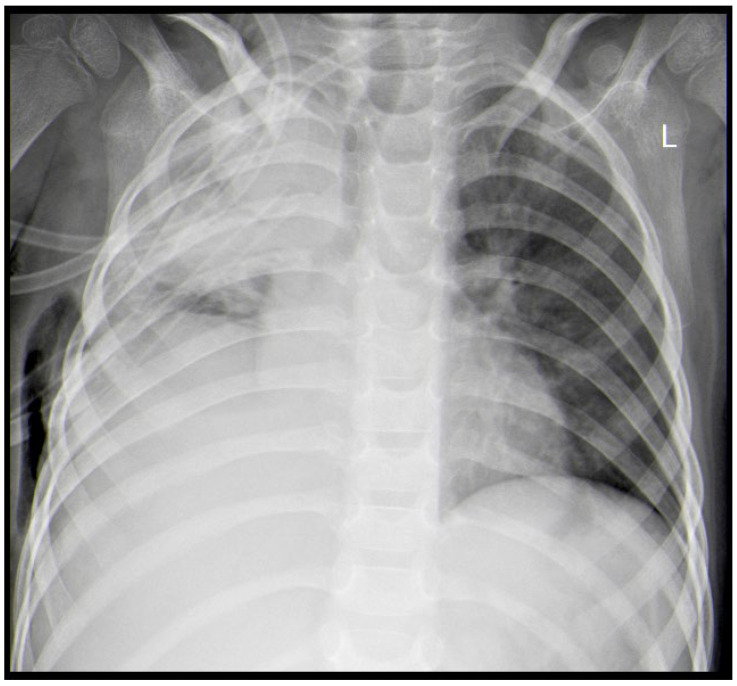
X-ray. Postoperative atelectasis (L—X-ray positive tag, marking patient’s left side).

**Figure 7 idr-18-00002-f007:**
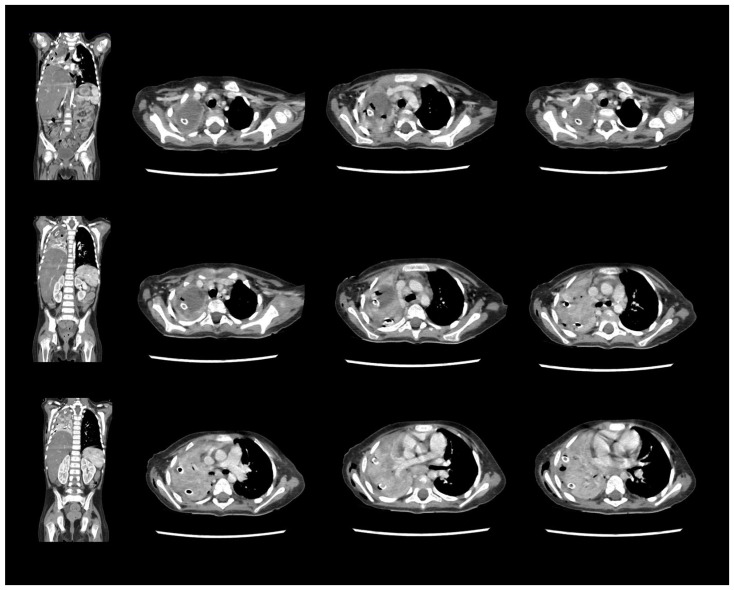
CT-scan. Enlargement of the inflammatory zone in the right upper pulmonary lobe.

**Figure 8 idr-18-00002-f008:**
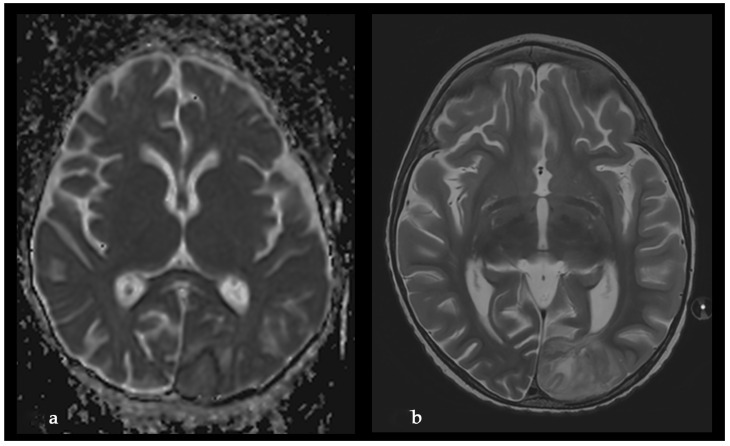
Initial MRI, before L-AmB treatment. (**a**) Reduced apparent diffusion coefficient on diffusion-weighted MRI. (**b**) MRI T2 sequence.

**Figure 9 idr-18-00002-f009:**
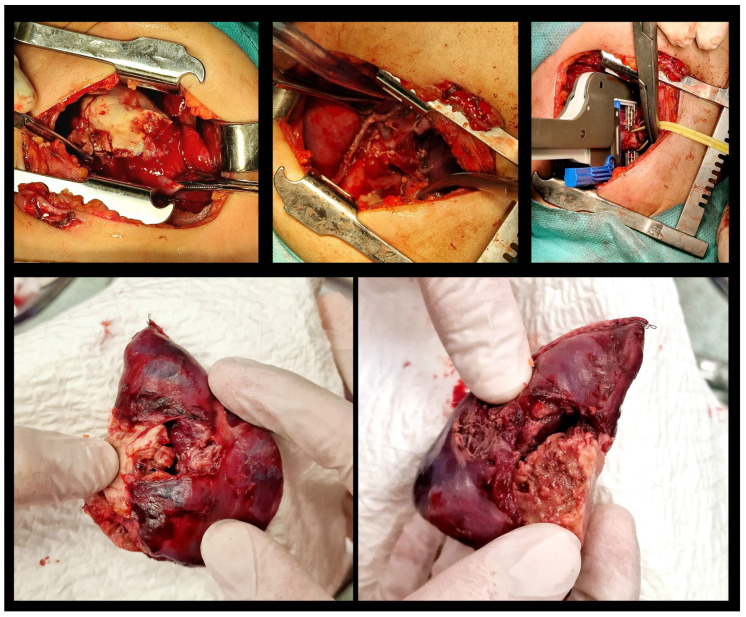
Right posteriolateral thoracotomy and right upper lobectomy. Surgical linear stapler bronchial resection.

**Figure 10 idr-18-00002-f010:**
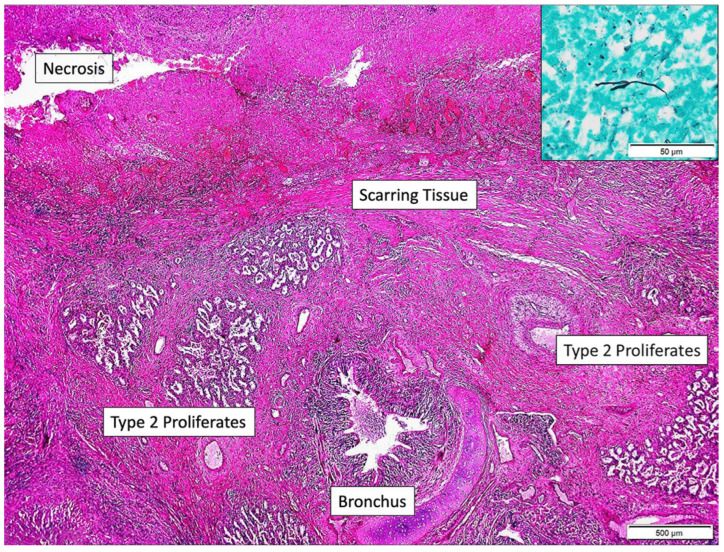
Mycotic granulomatous chronic pneumonia with extensive necrosis, associated with *Rhizomucor pusillus*. Positive hybridization for DNA of *Rh. pusillus*.

**Figure 11 idr-18-00002-f011:**
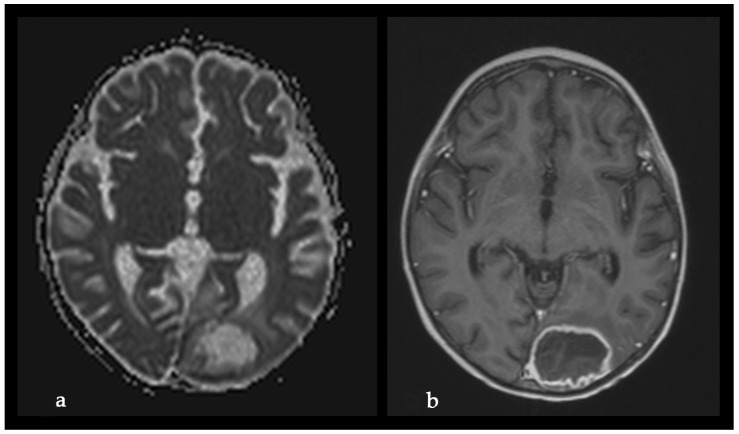
Second MRI, after L-AmB treatment, before neurosurgical treatment. (**a**) Absence of significant reduction in the apparent diffusion coefficient on diffusion-weighted MRI; (**b**) MRI T1 sequence with contrast enhancement: significant increase in the volume of the lesion and ring contrast enhancement.

**Figure 12 idr-18-00002-f012:**
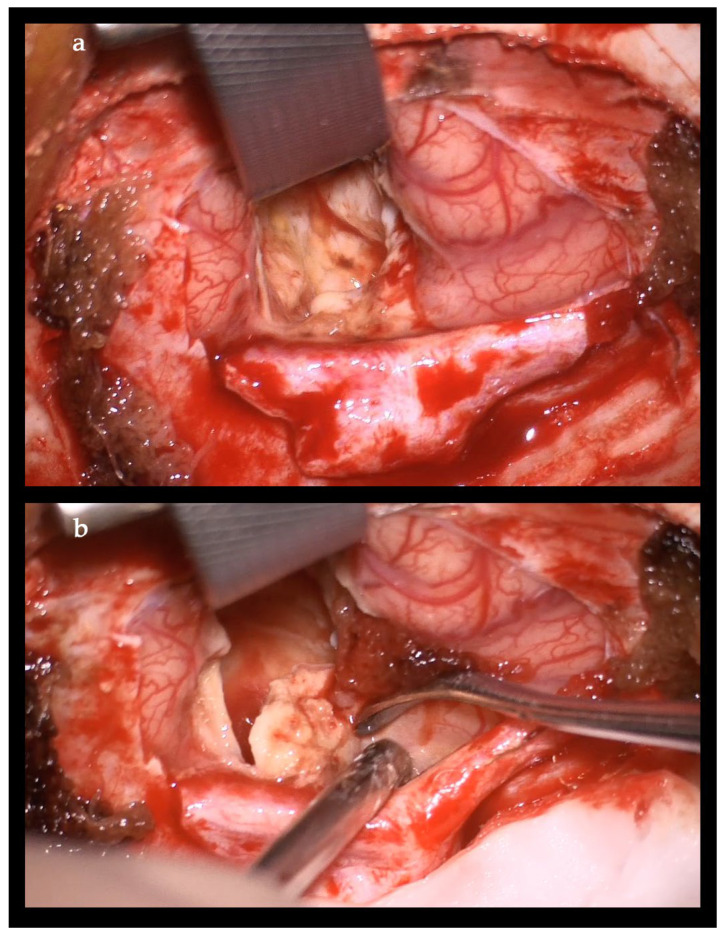
Intraoperative image during the evacuation of brain abscess. (**a**) Aspirable white abscess content; (**b**) dissection of the abscess capsula without a clear demarcation from the surrounding brain.

**Figure 13 idr-18-00002-f013:**
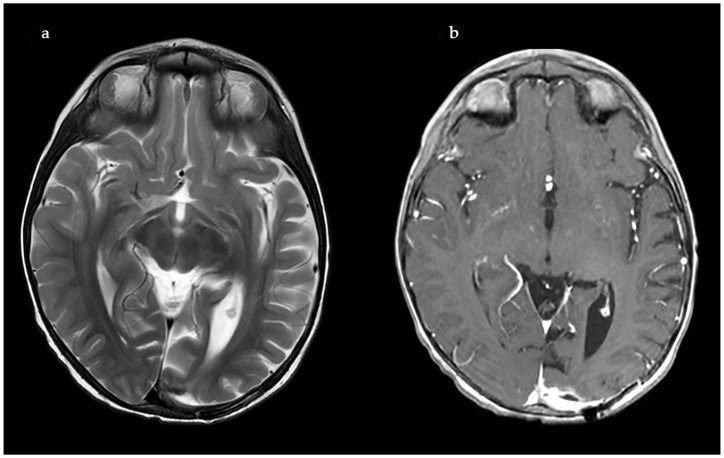
Postoperative MRI with complete shrinkage of the abscess cavity. (**a**) MRI T2 sequence; (**b**) MRI T1 sequence with contrast enhancement.

**Figure 14 idr-18-00002-f014:**
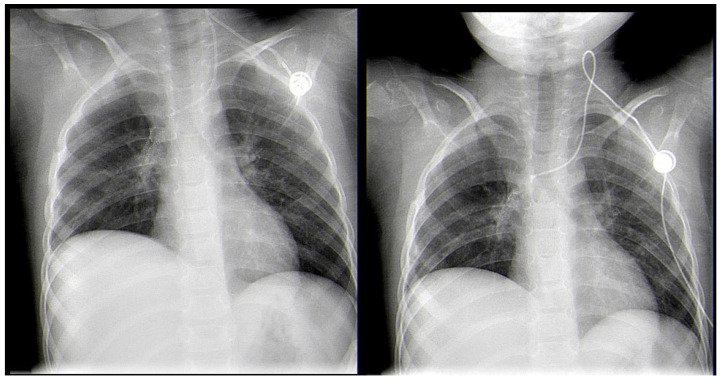
Late post-operative control X-rays with no signs of relapse of the lung lesions.

## Data Availability

The data presented in this study are available upon request from the corresponding author due to the GDPR protocols of the hospitals involved in the treatment of the reported patient.

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
