# Peer review of "A Rare Case of Rhizomucor pusillus Infection in a 3-Year-Old Child with Acute Lymphoblastic Leukemia, Presenting with Lung and Brain Abscesses—Case Report"

_2036-7449, 2025, doi:10.3390/idr18010002_

Round 1
Reviewer 1 Report
Comments and Suggestions for Authors
This is an interesting and clinically relevant case report of disseminated Rhizomucor pusillus infection in a pediatric patient with acute lymphoblastic leukemia, presenting with lung and brain abscesses. However, several essential issues should be addressed before acceptance.
Main concerns:
The text states: “One month after the onset of leukemia, during the chemotherapy-induced bone marrow aplasia”, but it remains unclear how many consecutive days the patient was neutropenic (<500/mm³).
The clinical course should explicitly state when chemotherapy was initiated in relation to COVID-19 infection, specifying whether the patient was symptomatic or asymptomatic at diagnosis, and whether positivity was confirmed in subsequent control samples (nasopharyngeal swabs or bronchoalveolar lavage [BAL]). Importantly, it is not mentioned whether the patient developed respiratory symptoms during follow-up.
Please detail the blood culture methodology available at your center. In the report it is written: “In the first blood culture test Candida spp. and coagulase-negative Staphylococci (CoNS) were confirmed.” It would be valuable to know whether this involved automated continuous-monitoring systems, species identification methods, and antifungal susceptibility testing.
The text describes persistence of fever and clinical symptoms. A graphical representation would be highly informative, showing at least daily body temperature, ANC (absolute neutrophil count), and CRP levels, together with therapeutic changes. This would allow the reader to better appreciate the patient’s evolution.
Please describe in more detail the fungal dissemination workup performed once candidemia was identified, and specify the clinical focus (e.g., CLABSI vs. another source).
Regarding the BAL: please indicate which microbiological assays were performed (culture, staining, specific PCR assays, and/or panfungal PCR).
It would be useful to discuss the potential role of Fungichip or other array-based techniques in your diagnostic approach.
The result of the brain biopsy must be indicated, and which microbiological assays were performed (culture, staining, specific PCR assays, and/or panfungal PCR).
Minor concerns:
Correct English
"The child showed no other neurological symptoms after that accident and was transferred to the tertiary pediatric hospital for further diagnosis."
"After 3 days under general anesthesia, an ultrasound guided percutaneous fine needle aspiration of the lung abscess was performed Visual diagnostics shows exudation, cavity, ground glass lesion, consolidation, pleural effusion, halo sign, atelectasis, reverse halo sign etc. [7,16]"
Comments on the Quality of English LanguageSeveral sentences required revision: “The child showed no other neurological symptoms after that accident” should be revised to “The child presented no additional neurological symptoms after the accident”
Changes words: Tenses shift irregularly, and terminology is sometimes inconsistent (e.g., “puss” instead of “pus,” “Pozaconazole” instead of “Posaconazole”).
Author Response
Dear Reviewer,
Thank you for the positive comments and remarks on our article. I totally agree with the following statements:
The minor language concerns were taken in consideration and a total language review was performed.
Comment 1: The text states: “One month after the onset of leukemia, during the chemotherapy-induced bone marrow aplasia”, but it remains unclear how many consecutive days the patient was neutropenic (<500/mm³).
Response1: added the period of neutropenia in lines 74-75 from the revised version
Comment 2: The clinical course should explicitly state when chemotherapy was initiated in relation to COVID-19 infection, specifying whether the patient was symptomatic or asymptomatic at diagnosis, and whether positivity was confirmed in subsequent control samples (nasopharyngeal swabs or bronchoalveolar lavage [BAL]). Importantly, it is not mentioned whether the patient developed respiratory symptoms during follow-up.
Response 2: Added day after chemotherapy for COVID 19 on line 87. The patient was asymptomatic added on line 88. Added the modality of subsequent negative COVID-19 tests in lines 88-89. In the follow-up added additional information - lines 237-240
Comment 3: Please detail the blood culture methodology available at your center. In the report it is written: “In the first blood culture test Candida spp. and coagulase-negative Staphylococci (CoNS) were confirmed.” It would be valuable to know whether this involved automated continuous-monitoring systems, species identification methods, and antifungal susceptibility testing.
Response 3: In lines 140-141 added Biofire FilmArray and other available methods in our clinic. However, the other verification methods from Hannover’s clinic of prof. Kreipe are clarified in a separate part of the article.
Comment 4: The text describes persistence of fever and clinical symptoms. A graphical representation would be highly informative, showing at least daily body temperature, ANC (absolute neutrophil count), and CRP levels, together with therapeutic changes. This would allow the reader to better appreciate the patient’s evolution.
Response 4: A figure (fig.3) with time distribution of symptoms with
CRP, Neu and fever added. The rest of the figures are rearranged accordingly. Therapeutic changes are described chronologically in the rest of the case report description.
Comment 5: Please describe in more detail the fungal dissemination workup performed once candidemia was identified, and specify the clinical focus (e.g., CLABSI vs. another source).
Response 5: the same as comment 3. We were aware of CLABSI in our case therefore removed the port-catheters and central IV-catheters and performed microbiological evaluations. Added lines 122-124
Comment 6: Regarding the BAL: please indicate which microbiological assays were performed (culture, staining, specific PCR assays, and/or panfungal PCR).
Response 6: same as response 3
Comment 7: It would be useful to discuss the potential role of Fungichip or other array-based techniques in your diagnostic approach.
Response 7: Fungichip was used by Prof.Kreipe to confirm the uncertain diagnosis. It is out of my competence to comment in details the method. if needed I will contact directly the Hannover clinic and will discuss the topic in further rounds.
Comment 8: The result of the brain biopsy must be indicated, and which microbiological assays were performed (culture, staining, specific PCR assays, and/or panfungal PCR).
Response 8: Added statement in lines 222-224
Reviewer 2 Report
Comments and Suggestions for Authors
The case report presented by Pahnev et al is very extensively documented with high quality figures. The case report presents a very rare disease Invasive Mucormycosis (IM) caused by Rhizomucor pusillus. IM is a life threatening disease with high mortality rate. The prognosis is often poor and is treated by antifungal Amphotericin B. The case report is well written and clinically relevant. Here are my comments:
1) How was the infecting organism identified in this case report? Please explain
2) The antifungal regimen is described briefly. Please clarify
3) The patient outcome is critical to understanding management success. Please provide any follow up details and the status of the diagnostic.
4) All species name need to be italicized. For example, line 94, Candida needs to be italicized.
5) The novelty of this case is asserted but not demonstrated. The authors should summarize prior reports of pediatric Rhizomucor infection (especially with CNS involvement) and highlight distinguishing features of this case.
Author Response
Dear Reviewer,
Thank you for the positive comments and remarks on our article. I totally agree with the following statements:
Comment 1: How was the infecting organism identified in this case report? Please explain
Response1: Added Biofire FilmArray, available at our center (line 140 of the revised version)
Comment 2: The antifungal regimen is described briefly. Please clarify
Response 2: There is no widely accepted protocols for treatment of these type of infections especially in children. However, the applied therapy was precised according to the outcomes form the microbiological evaluations of the materials acquired from blood cultures and specimens from surgeries. The final therapy was accepted after the DNA-verification of Rh.pusillus. (Line 230)
Comment 3: The patient outcome is critical to understanding management success. Please provide any follow up details and the status of the diagnostic.
Response 3: Extended follow-up text in the “Case report” part (line 236 and following)
Comment 4: All species name need to be italicized. For example, line 94, Candida needs to be italicized.
Response 4: Corrected.
Comment 5: The novelty of this case is asserted but not demonstrated. The authors should summarize prior reports of pediatric Rhizomucor infection (especially with CNS involvement) and highlight distinguishing features of this case.
Response 5: Added discussion in the literature (lines 244-248)
Reviewer 3 Report
Comments and Suggestions for Authors
This is a relevant case that deserves communication, in particular for the therapeutic approach used by the authors and the outcome. The case is well presented, and there are only some minor issues to be solved:
Starting from the title, scientific names must be given in italics, genus starts with an upper case and species with a lower case. Use this format throughout the whole manuscript.
HM is used as an abbreviation in the abstract, but the full concept is given afterwards.
How was active tuberculosis discarded?
Bacteria and fungi were isolated from blood and other tissues. How were they identified at the species level?
SARS-CoV-2 is not diagnosed by PCR; it is an RNA virus.
The meaning of the sentence given in line 137 is confusing.
There is no explanation of what is observed in the insert of Figure 9.
Author Response
Dear Reviewer,
Thank you for the positive comments and remarks on our article. I totally agree with the following statements:
Corrected tittle and body fonts and italics.
Comment 1: HM is used as an abbreviation in the abstract, but the full concept is given afterwards.
Response1: The abbreviation is explained in the abstract body.
Comment 2: How was active tuberculosis discarded?
Response 2: Added TB-SPOT and Quatiferon modalities of IGRAs (line 94-95 in the revised version)
Comment 3: Bacteria and fungi were isolated from blood and other tissues. How were they identified at the species level?
Response 3: Added Biofire FilmArray, available at our center (line 140)
Comment 4: SARS-CoV-2 is not diagnosed by PCR; it is an RNA virus.
Response 4: Corrected to RT-PCR
Comment 5: The meaning of the sentence given in line 137 is confusing.
Response 5: The sentence divided to 2 more clear sentences
Comment 6: There is no explanation of what is observed in the insert of Figure 9.
Response 6: The fig.9 is now fig. 10. Added time distribution graphic for the symptoms. The provided picture is from prof.Kreipe and it contains both microscopy and positive DNA hybridization in upper right corner.
Round 2
Reviewer 1 Report
Comments and Suggestions for Authors
I thank the authors for appropriately addressing the suggestions.
Author Response
Dear Reviewer,
Thank you very much for the supportive and encouraging comments on our work.
Looking forward for future collaborations
Best regards,
Yanko Pahnev
Reviewer 2 Report
Comments and Suggestions for Authors
The authors have answered to all my queries satisfactorily.
Author Response
Dear Reviewer,
Thank you very much for your constructive, supportive and encouraging comments on our work.
Looking forward for future collaborations
Best regards,
Yanko Pahnev